

# MDAS: A New Multimodal Benchmark Dataset for Remote Sensing

Jingliang Hu[1], Rong Liu[1], Danfeng Hong[2], Andrés Camero[2], Jing Yao[3], Mathias Schneider[2],
Franz Kurz[2], Karl Segl[4], and Xiao Xiang Zhu[1,2]

[1]Data Science in Earth Observation (SiPEO), Technical University of Munich (TUM), 80333 Munich, Germany
[2]Remote Sensing Technology Institute (IMF), German Aerospace Center (DLR), 82234 Wessling, Germany
[3]Key Laboratory of Digital Earth Science, Aerospace Information Research Institute, Chinese Academy of Sciences, Beijing 100094, China
[4]German Research Center for Geosciences (GFZ), Helmholtz Center Potsdam, Telegrafenberg A17, 14473 Potsdam, Germany

**Correspondence:** Xiao Xiang Zhu (xiaoxiang.zhu@dlr.de)

**Abstract.** In Earth observation, multimodal data fusion is an intuitive strategy to break the limitation of individual data. Complementary physical contents of data sources allow comprehensive and precise information retrieve. With current satellite missions, such as ESA Copernicus programme, various data will be accessible at an affordable cost. Future applications will have many options on data sources. Such privilege can be beneficial only if algorithms are ready to work with various data

sources. However, current data fusion studies mostly focus on the fusion of two data sources. There are two reasons, first, different combinations of data sources face different scientific challenges. For example, the fusion of synthetic aperture radar (SAR) data and optical images needs to handle the geometric difference, while the fusion of hyperspectral and multispectral images deals with different resolutions on spatial and spectral domains. Second, nowadays, it is still both financially and labour expensive to acquire multiple data sources for the same region at the same time. In this paper, we provide the community a

benchmark multimodal data set, MDAS, for the city of Augsburg, Germany. MDAS includes synthetic aperture radar (SAR) data, multispectral image, hyperspectral image, digital surface model (DSM), and geographic information system (GIS) data. All these data are collected on the same date, $7^{th}$ May 2018. MDAS is a new benchmark data set that provides researchers rich options on data selections. In this paper, we run experiments for three typical remote sensing applications, namely, resolution enhancement, spectral unmixing, and land cover classification, on MDAS data set. Our experiments demonstrate the

performance of representative state-of-the-art algorithms whose outcomes can sever as baselines for further studies.

## 1 Introduction

The rapid development of Earth observation (EO) techniques enables the measurements and monitoring of Earth on the land surface and beneath, of the quality of air and water, and of the health of humans, plants, and animals. Remote sensing (RS) is one of the most important contact-free sensing means for EO to extract relevant information about the physical properties of the Earth and environment system from space. With the ever-growing availability of RS data source from both satellite

and airborne sensors on a large scale and even global scale, e.g., Sentinel missions, multi-modal RS techniques have been garnering increasing attention in various EO-related tasks (Hong et al. 2021c). The data acquired by different platforms can provide diverse and complementary information, such as light detection and ranging (LiDAR) providing the height information



about the ground elevation, synthetic aperture radar (SAR) providing the structure information about Earth's surface, and
multispectral (MS) or hyperspectral (HS) data providing detailed content information of sensed materials. The joint exploitation
of different RS data has been therefore proven to be helpful to further enhance our understanding, possibilities, and capabilities
to Earth and environment.

Multimodal data fusion takes advantage of complementary information of different data sources so that its performance can
reach beyond the limitation of individual data. Currently, a lot of studies in remote sensing focus on developing methodologies
that can effectively utilize different data sources. Tupin and Roux (2003); Hu et al. (2019); Hong et al. (2020c); Meraner
et al. (2020); Adrian et al. (2021); Hong et al. (2021f) develop algorithms to join advantages of optical images and synthetic
aperture radar (SAR) data for their respective goals. They utilize spatial structures, spectral information from optical data and
the independence of weather conditions, dielectric properties, geometric structures from SAR data. Paris and Bruzzone (2014);
Khodadadzadeh et al. (2015); Al-Najjar et al. (2019); Hang et al. (2020); Ge et al. (2021); Hong et al. (2020a) introduce
models to merge height information from LiDAR data or digital surface model (DSM) with spatial and spectral information
from optical images. Yokoya et al. (2011); Simoes et al. (2014); Hong et al. (2019b, c); Zhang et al. (2020c); Liu et al. (2020)
propose methods to fuse rich spectral information of hyperspectral images with high spatial resolution of multispectral images.
All these studies have proven that the data fusion is very beneficial to remote sensing applications.

While reviewing literature of data fusion, one can find that some data sets frequently appear in a lot of studies. For example,
the hyperspectral image of Houston university[1]; the aerial images and DSM of Vaihingen (Rottensteiner et al. 2012). These
accessible benchmark data sets play a very important role in the advancing of fusion technologies. Especially, in this era, data
are the foundation for state-of-the-art machine learning and deep learning methodologies. As remote sensing data strongly
associate to geographic locations, it would be beneficial to the community to have more benchmark data sets of high quality
that origin from different geolocations.

In the remote sensing community, benchmark data sets are often generated based on one data source, such as the UC-
Merced dataset (Yang and Newsam 2010), the WHU-RS19 dataset (Xia et al. 2010), the AID dataset (Xia et al. 2017), and the
BigEarthNet (Sumbul et al. 2019). Recently some researchers create benchmark data sets for multimodal remote sensing data,
such as Hong et al. (2021e), including hyperspectral and multispectral data, hyperspectral and SAR data, hyperspectral, SAR,
and DSM data. Besides, the IEEE GRSS data fusion contests have done great contributions by consistently publishing data sets
for developing fusion technologies. They often provide two to three types of data for tasks like classification, detection, and 3D
reconstruction (Wu et al. 2019, 2020). By far, the listed datasets mostly have a comparative large amount of samples and wide
geographic distribution. Because it is easy to access the original data sources, such as Sentinel-1, Sentinel-2, and Google Earth
images. When involving certain data sources, namely hyperspectral image, LiDAR data, and DSM, the amount of samples is
often limited due to the high costs of collections and data preparations. Examples are the Vaihingen dataset and the Houston
dataset. The complexity of preparing these data is also one of the reasons that most studies focus on fusing two data sources,
often the easily accessible ones.

---

[1]https://hyperspectral.ee.uh.edu/?page_id=1075





In this paper, we contribute to the community by publishing a benchmark remote sensing dataset including five modalities, namely, SAR data, multispectral image, hyperspectral image, DSM, and GIS data, for the entire city of Augsburg, Germany. Apart from the GIS data, the remote sensing data are all collected on the same day. The preparations of the SAR data, the hy-
perspectral image, and the DSM are carried out by a group of experienced experts and ensure the high quality. We also provide experiment results of state-of-the-art algorithms in this paper for three typical remote sensing tasks, namely, resolution enhancement, spectral unmixing, and land cover classification. These results play as baselines for future studies. MDAS not only provides one more testing instance for developing algorithms of typical tasks, but also possibilities for unknown applications because of the variety of data modalities.

## 65   1.1   Super-resolution

Images with high spatial resolution have detailed textures and structures that facilitate semantic interpretation. Meanwhile, materials have different reflectances at different ranges of wavelengths, a high spectral resolution and a wide spectral range can benefit the identification of covering material. However, the limitation of imaging sensors prevent achieving high resolution in spectral and spatial domains at the same time (Loncan et al. 2015; Hong et al. 2021g). To break the limitation, researchers
develop super-resolution algorithms to fuse data of the two types so that high resolution in both spatial and spectral domains are available. With the increasing amount of multispectral and hyperspectral data, e.g. Sentinel-2 images and EnMAP images, the super-resolution products have an important role in optical remote sensing.

Current super resolution research (Sheikholeslami et al. 2020; Yokoya et al. 2017; Zhang et al. 2020a, b) evaluates algorithms based on data sets which have high spatial resolutions. They down-sample the spatial resolution of original data sets to create
the counterpart low resolution data. On spectral domain, they mostly apply either the same spectral bands or a subset of the spectral bands. Very few would utilize simple spectral response functions. These data configurations simplify the super resolution applications of the real world. On this regard, in MDAS, we attempt to provide realistic data sets. We simulate EnMAP and Sentinel-2 images using the EeteS (Segl et al. 2012, 2010) and S2eteS (Segl et al. 2015) software based on a HySpex imagery. This simulation integrates not only the specific spatial response functions and spectral response functions, but
also the effects of instrumental and environmental parameters on the resulting image characteristics. Besides the simulations, we also provide a Sentinel-2 image from the same date for the same region.

When comparing to the current data sets used for super resolution evaluations that mostly only emphasize the spatial resolution, MDAS also challenges the algorithms with respect to spectral enhancement, instrumental effects, environmental impact, and influence of different sensors.

## 85   1.2   Spectral unmixing

Hyperspectral remote sensing has greatly improved our ability to qualitatively and quantitatively sense Earth's surface, due to its capability to obtain data with tens to hundreds of contiguous spectral bands (Hong et al. 2020b). However, mixed pixels are widely spread in hyperspectral image due to its limited spatial resolution, which has hindered the accurate analysis and applications of hyperspectral data. Hyperspectral unmixing is one of the most hot topics in the hyperspectral image processing,



which aims at solving the mixed pixel problem by decomposing the mixed pixel into the constituent pure signatures, also called as endmembers, and the corresponding abundance coefficients (Liu et al. 2017; Hong et al. 2021d).

The unmixing topic has been intensively studied by researchers in the last few decades, and many efforts have been made to improve the accuracy of the methods of unmixing. The most widely used real hyperspectral image datasets for the evaluation of the developed unmixing algorithms were presented by Zhu (2017), in which the Urban dataset and Cuprite dataset are exceedingly frequently used. These images are small in spatial size, which is not enough to test whether the developed method is capable of processing large dataset with low error and reasonable time cost. Hong and Zhu proposed a novel subspace unmixing framework with the use of low-rank attribute embedding, called SULoRA (Hong and Zhu 2018). Moreover, they further considered the spectral variability and proposed a seminal linear mixing model, called augmented linear mixing model (ALMM), for accurate spectral unmixing (Hong et al. 2019a). In addition, the ground truth of abundance is usually not available and only synthetic images can be used for the quantitative evaluation of the abundance (Du et al. 2019). However, synthetic images can't model the complex scene of real images. Thus, large dataset with reliable reference is highly requested for the development of efficient unmixing methods.

### 1.3 Multimodal land cover classification

Land cover classification has been a fundamental but challenging research topic in the RS community, as many high-level subsequent analysis largely depends on the classification results. Over the past decades, extensive classification algorithms were developed for single modalities (Huang et al. 2020; Hong et al. 2021b; Rasti et al. 2020), e.g., HS, MS, SAR, LiDAR, OpenStreetMap. These methods have been proven to be effective in many real applications. We have to admit, however, that the utilization of single modalities inevitably suffers from the performance bottleneck. Enormous efforts have been recently made to couple or jointly analyse different RS observation sources by the attempts to design advanced data fusion methods to achieve a more diversified description for a studied scene. According to different processing levels that the fusion behaviour happens, the current state-of-the-art approaches related to RS data fusion can be approximately categorized into three groups: *Data Level*, *Feature Level*, and *Decision Level*.

The first application of this paper, i.e., HS super-resolution, is a typical *Data Level* fusion task. These developed super-resolution approaches are only applied for homogeneous RS data fusion, e.g., HS and MS images, panchromatic and MS images. Because they have similar imaging principle and data structure, enabling the fusion of these data sources. For heterogeneous RS data, e.g., SAR and optical data, DSM and SAR, or DSM and optical data, etc., they fail to directly fuse in the image level due to their totally different image properties and structure. For this reason, the *Feature Level* fusion strategy has been garnering increasing attention, providing greater potential in the land cover classification task. Very recently, only few benchmark datasets are openly available for multimodal RS data processing and analysis, e.g., land cover classification. More notably, there are some potential limitations in these existing and public benchmark datasets. For example,



- the number of multimodal RS data are limited, e.g., the well-known Houston 2013 datasets (including HS and DSM data) provided by the IEEE Data Fusion Contest (DFC) 2013, the LCZ datasets (including MS and OpenStreetMap) provided by the IEEE DFC 2017, the optical and LiDAR data provided by the IEEE DFC 2019, etc.;

- there is a lack of diverse multimodal data, especially heterogeneous data, e.g., the Houston 2018 datasets provided by
the IEEE DFC 2018, including HS, MS, RBG.

For these reasons, building high-quality and multimodal RS benchmark datasets is a primary and key step to boost the development of various multimodal RS applications, further contributing to the RS community. This, therefore, will motivate us to develop new and more diversified multimodal RS datasets.

### 1.4 Contribution of this paper

In this paper, we have two major contributions. First, we publish MDAS, a new multimodal benchmark data, that consists of five modalities. All components of MDAS are well-prepared by experienced experts to ensure a high quality, e.g. the preparation of the DLR 3K DSM, the simulation of EnMAP imagery, and the generation of ground reference for unmixing. As far as we know, the ground reference is the first product that provides reliable reference for quantitative evaluation of real hyperspectral image unmixing. Besides the high quality, the variety of data modality provides more possibilities for data fusion applications.
Second, we demonstrate the performance of state-of-the-art algorithms for three typical remote sensing applications, namely, resolution enhancement, spectral unmixing, and land cover classification. These experiment results provide baselines for further algorithm developments on MDAS.

## 2 MDAS dataset

MDAS has remote sensing data of five modalities. They are SAR data, multispectral image, hyperspectral image, DSM, and
140 GIS data. Fig. 1 demonstrates all these modalities. The region of interest (ROI) is the city of Augsburg, Germany, and covers an area of 121.7 $km^2$. Apart from the GIS data, the other data are all collected on the same day, $7^{th}$ May 2018.

### 2.1 Synthetic aperture radar data

The SAR component in MDAS is a Sentinel-1 data. We download a level-1 Ground Range Detected (GRD) product collected under the Interferometric Wide (IW) Swath mode. It is a dual-Pol SAR data with VV and VH channels. Based on recommended
preprocessing of Sentinel-1 data in Hu et al. (2018); Filipponi (2019); Zhu et al. (2020), we prepare the SAR data by using ESA SNAP toolbox[2] following the workflow of applying precise orbit profile, conducting radiometric calibration, and performing terrain correction. The topographic data used in the processing is the Shuttle Radar Topography Mission (SRTM). We apply bilinear interpolation to achieve a geocoded SAR image with a 10 meter ground sampling distance (GSD). Based on the ROI,

---

[2]https://step.esa.int/main/toolboxes/snap/

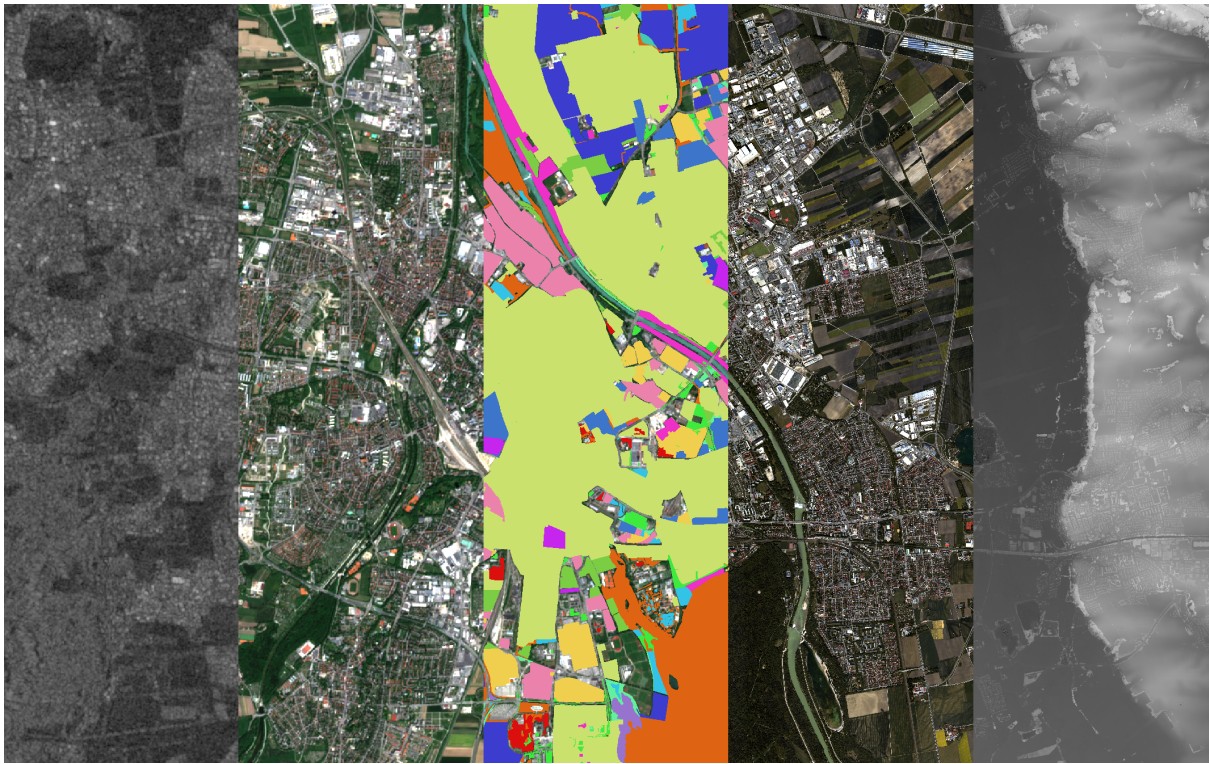

**Figure 1.** The Visualization of five data modalities in MDAS. From left to right: the Sentinel-1 image (VH band in dB), the Sentinel-2 image (RGB: 665 nm, 560 nm, 490 nm), the OSM land use layer, the HySpex image (RGB: 691 nm, 569 nm, 445 nm), and the DSM.

the final SAR image has a size of $1371 \times 888$ pixels, a geographic reference of WGS 84/UTM Zone 32 N, and two channels
(intensities of VV and VH).

## 2.2 Multispectral imagery

The multispectral imagery in MDAS is a Sentinel-2 data. We fetch a level-2A product directly from the Sentinel Data Hub. It is an image in Bottom Of Atmosphere (BOA) reflectance and geocoded in WGS 84/UTM Zone 32 N as well. It has 12 spectral bands with wavelength ranging from 440 to 2200 nm. The six bands (B5, B6, B7, B8A, B11, B12) of 20 meter GSD and the
two bands (B1, B09) of 60 meter GSD are all upsampled to 10 meter GSD. The final image is cropped according to the ROI, resulting in a size of $1371 \times 888$ pixels.



### 2.3 Hyperspectral imagery

#### 2.3.1 Data acquisition

The hyperspectral images used in this study have been acquired using the airborne imaging spectrometer system HySpex,
which is operated by the Remote Sensing Technology Institute (IMF) of the German Aerospace Center (DLR). For this study,
HySpex was mounted on a Dornier DO228-212. The imagery over the city of Augsburg was acquired in 23 flight strips.

#### 2.3.2 System configuration

The system features two different cameras covering the VNIR and SWIR spectral domain. Both cameras have been extensively
characterized at the Calibration Home Base (CHB, Brachmann et al. 2016), resulting in a very well characterized high precision
instrument suited for benchmark earth observation applications. The HySpex VNIR-1600 features a CCD detector covering the
spectral range 416 to 992 nm with 160 channels. This results in a spectral sampling interval of 3.6 nm. The spectral resolution
ranges from 3.5 nm at nadir to approximately 6 nm at the outer edge of the swath. The HySpex SWIR-320m-e is equipped
with a mercury cadmium telluride (MCT) detector with 256 channels covering the spectral range 968 to 2498 nm at a sampling
interval of 6 nm and a spectral resolution of 5.6 to 7.0 nm. The VNIR detector has a width of 1600 pixels, while the SWIR
detector has a width of 320 pixels (Köhler 2016).

#### 2.3.3 Preprocessing

The HySpex processing chain consists of two parts: The preparation of the data for the automatic processing which comprises
the processing of GPS and IMU data and assigning them to the corresponding flight strips, and the second part, consisting
of systematic correction, orthorectification, co-registration and atmospheric correction which is performed in the generic pro-
cessing system Catena (Krauß et al. 2013). For the systematic correction, the following steps are applied on each frame in the
given order: Dark signal correction, linearity correction (VNIR only), stray light correction (VNIR only), radiometric calibra-
tion, bad pixel correction (SWIR only) and finally correction of point spread function (PSF) non-uniformities. The linearity
correction of the VNIR instrument is performed after the dark signal correction, since the dark current is very stable and does
not significantly change. For the stray light correction, a four dimensional tensor is applied. Currently, only the VNIR data
are corrected for stray light, while the measurement of the SWIR instrument's stray light is currently ongoing research. The
radiometric calibration is performed by dividing the signal corrected to this point by the radiometric response multiplied by the
integration time. The radiometric responses of both instruments are determined during laboratory calibration. The bad pixels
of the HySpex SWIR are corrected by linear interpolation between adjacent pixels along the spectral axis, while the bad pixels
of the VNIR are corrected by the camera software. Finally, optical distortions like smile, keystone and - equally important -
the individual angular and spectral resolutions of each pixel are corrected by applying an individual homogenization kernel
to each pixel. For the co-registration of VNIR and SWIR data, a BRISK matching is used (Schwind et al. 2014). After this
step, the data are orthorectified using the physical sensor model, the GPS-/IMU-data, the mounting angles and the DEM by the





DLR software ORTHO, followed by an atmospheric correction using the DLR software ATCOR (Richter 1998). As reflectance depends on the viewing and solar illumination geometry, the processed data strips appear inconsistently in magnitudes, which

results in different brightness. This phenomenon is described by the bidirectional reflectance distribution function (BRDF). We manually iterate a BRDF effects correction routine (BREFCOR, Schläpfer et al. 2014) on our imagery and dramatically mitigate the influence of BRDF. At last, the data strips are merged as one scene for the entire city of Augsburg.

### 2.3.4  Simulation of additional image products

The hyperspectral Augsburg mosaic is an ideal data basis to derive further image products suitable for classification tests or

195 resolution enhancement experiments. Accordingly, very realistic hyperspectral EnMAP and multispectral Sentinel-2 data were generated. This was done using the EeteS (Segl et al. 2012, 2010) and S2eteS (Segl et al. 2015) software, both developed within the EnMAP project (Guanter et al. 2015) to understand better the effects of instrumental and environmental parameters on the resulting image characteristics. The sequential processing chain of these software tools consists of four independent parts - the atmospheric, spatial, spectral and radiometric modules, in which the HySpex BOA reflectance data is used to calculate

top-of-atmosphere radiance and subsequently digital numbers. This forward simulator is coupled with a backward simulation branch consisting of calibration modules (e.g. for EnMAP: non-linearity, dark current and absolute radiometric calibration) and a series of pre-processing modules (radiometric calibration, co-registration, atmospheric correction and orthorectification) forming the complete end-to-end simulation tool. Since both image products are based on the same data, they are ideally suited for comparisons of classifications or parameter retrievals using EnMAP and Sentinel-2 images. Additional three images were

generated for resolution experiments, which do not contain any additional sensor or processing uncertainties for better comparison. Accordingly, an end-to-end simulation was completely omitted here and only a purely spectral and spatial adaptation was calculated. In a first step, the HySpex data were spatially adjusted to 10 m and 30 m GSD respectively. This was done with the help of a PSF convolution using a 2D Gaussian filter, whose FWHM results from the ratio of the output and input GSD. Since the input grid (2.2 m grid size) and output grids (10.0/30.0 m) do not overlap perfectly, the result of the PSF convolution

is slightly erroneous because the PSF cannot be perfectly positioned over the input pixels. In order to keep these errors as small as possible, the PSF was spatially shifted in the Fourier space for each pixel and thus optimally adapted to the input grid. Subsequently, the 10 and 30 m GSD data were spectrally adjusted to the 242 EnMAP bands by a spectral filtering using the original spectral response functions of ENMAP. Similar, the four Sentinel-2 10 m bands were also generated based on the 10 m GSD data.

## 2.4  Digital Surface Model

A high resolution DSM with images acquired with the 3K camera system (Kurz et al. 2012) has been generated for this study (see example of sub area in figure 2). For this, 1702 images from the nadir looking camera, which has an average coverage of $1170m \times 765m$ and a GSD of $25cm$, were used for the further processing. These images were acquired with 23 flight strips covering the whole test area of $275km^2$ size at a flight height of $1800m$ above ground, which results in a 75% and 36% along

respectively across track overlap. First, a self-calibration image bundle adjustment was performed using the SAPOS corrected

Data

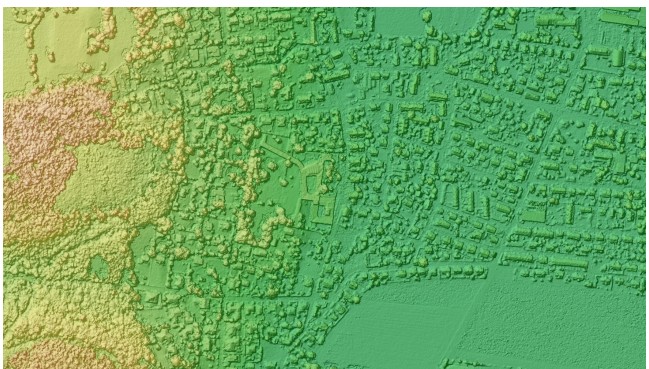

**Figure 2.** Shaded DSM of subarea.

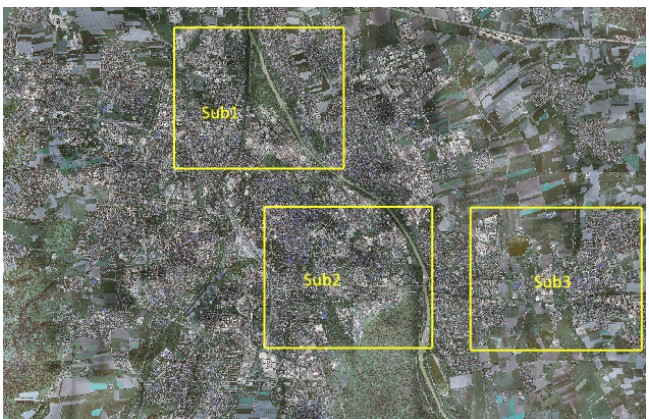

**Figure 3.** Three manually labelled subregions.

and filtered GNSS/inertial data of the image positions and attitudes as well as 121116 tie points. No further pass information was introduced except the global SRTM elevation model as a weak additional height information for the tie points. With precise oriented images, the DSM generation was then performed using the SGM algorithm with modifications (d'Angelo and Kurz 2019). The resulting disparity images for each stereo pair were filtered, fused, filled and finally resampled to a GSD of $30cm$.

**2.5 Geographic Information System (GIS) data**

For the GIS data in MDAS, we download the building layer, the land use layer, and the water layer from Open Street Map[3]. The original vector layers are rasterized into GeoTiff files with a coordinate system of WGS 84/UTM Zone 32 N. Depend on the application, the GIS data can be utilized as either input data or ground reference.

---

[3]https://www.openstreetmap.org/#map=13/48.3699/10.8809

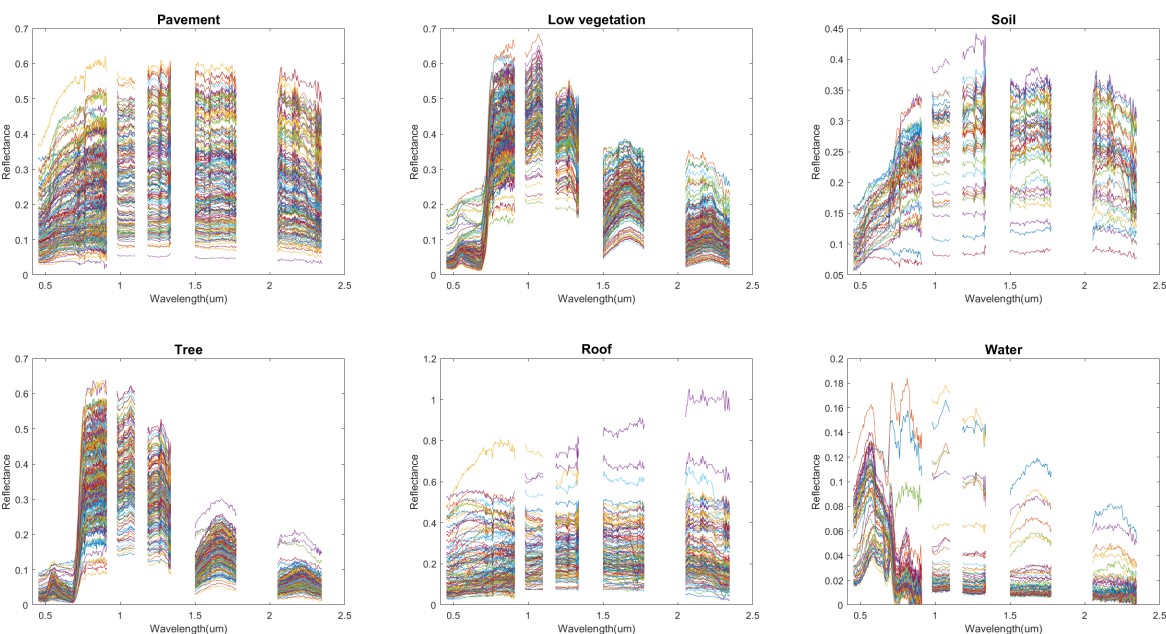

**Figure 4.** The endmember spectra for each land cover class.

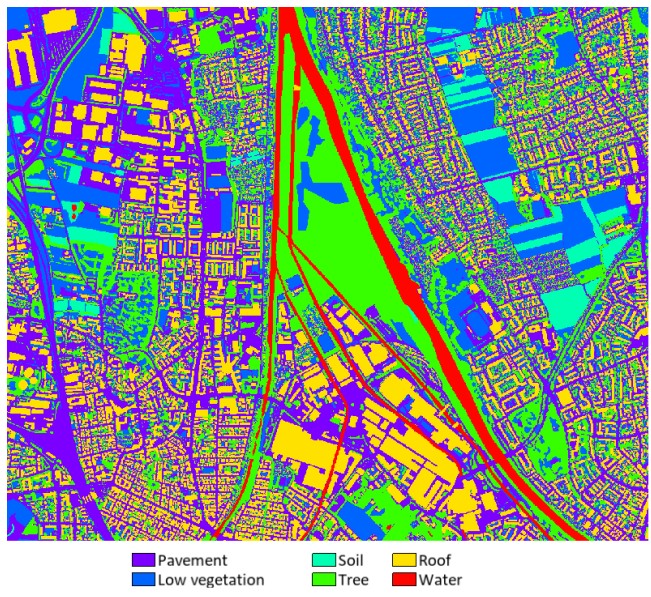

**Figure 5.** The manually delineated land cover composition of the sub1 region based on the DLR 3K image.

Earth System
Science
Data

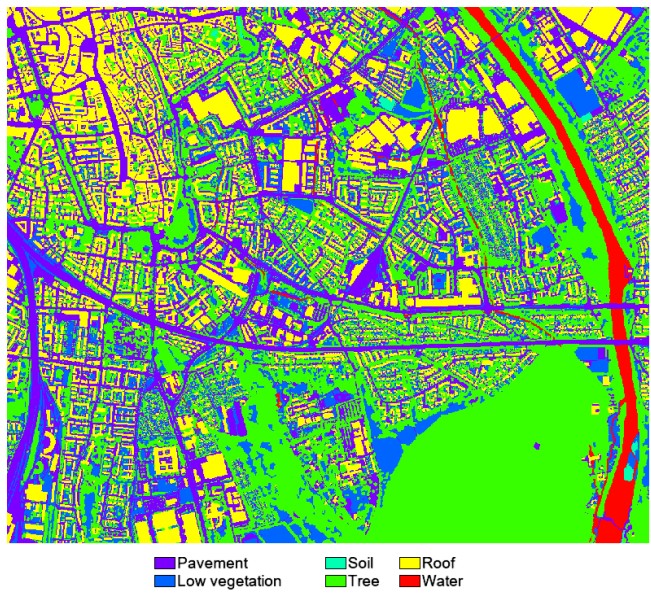

Pavement    Soil    Roof
Low vegetation    Tree    Water

**Figure 6.** The manually delineated land cover composition of the sub2 region based on the DLR 3K image.

## 2.6    Manually labeled data

Three subregions were selected to be labelled, and their location in the original image is shown in Fig. 3. A spectral library of the most dominant urban materials for the HySpex hyperspectral image was constructed by manually chosen pure pixels throughout the subregions of the HySpex data. In total, the spectral library contains 859 endmember spectra with 181 spectra being pavement, 211 spectra being low vegetation, 49 spectra being soil, 210 spectra being tree, 134 spectra being roof and 74 spectra being water. The spectra for each land cover class are shown in Fig. 4. To obtain sub-pixel level information of mixed

pixels for the unmixing task, the land cover mapping products of six classes, namely, pavement, low vegetation, soil, tree, roof and water, were manually delineated across the 0.2 m DLR 3K image. The elaborately produced land cover maps of the three subregions are presented in Fig. 5-7. The ground-truth maps can be used to generate reference abundances for quantitative evaluations of the hyperspectral unmixing task.

## 2.7    Data summary

The data components of MDAS are listed in Table 1. They are a Sentinel-1 SAR image, a Sentinel-2 multispectral image, a DLR 3K DSM, a HySpex hyperspectral image. These four components are data sets collected from operating missions. Besides, MDAS includes some HySpex-based simulated data. The S2eteS_S2 is a simulation of Sentinel-2 image in terms of the spatial and spectral characteristics. The EeteS_EnMAP_10 imitates the spectral bands of EnMAP yet with a 10 meter GSD. The EeteS_EnMAP_30 imitates both the spectral bands and 30 meters GSD of EnMAP. Meanwhile, the simulation of EnMAP

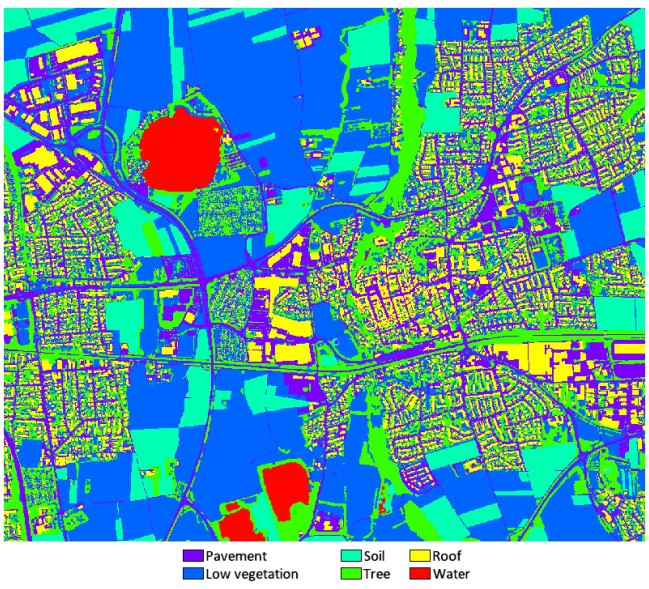

**Figure 7.** The manually delineated land cover composition of the sub3 region based on the DLR 3K image.

**Table 1.** This table lists data components of MDAS.

| Data | Modality | Sensor |
|---|---|---|
| Sentinel-1 | SAR | Sentinel-1 payload |
| Sentinel-2 | Multispectral image | Sentinel-2 payload |
| DSM | DSM | DLR 3K |
| HySpex | Hyperspectral image | HySpex |
| S2eteS_S2 | Multispectral image | S2eteS Spatial and spectral simulation |
| EeteS_EnMAP_10 | Hyperspectral image | EeteS Spatial and spectral simulation |
| EeteS_EnMAP_30 | Hyperspectral image | EeteS Spatial and spectral simulation |
| EnMAP | Hyperspectral image | EeteS |
| GIS | GIS | Open street map (OSM) |
| Endmember | | Manual labeling |
| Land cover maps | | Manual labeling |

includes additionally the effects of instrumental and environmental parameters. At last, MDAS also has components which can be used as ground reference. They are OSM GIS maps and manually labelled endmember and land cover maps.

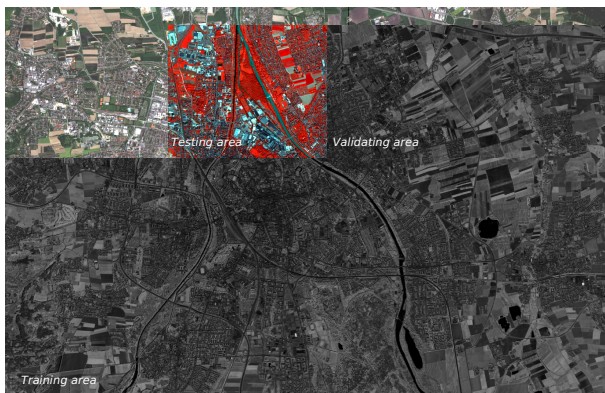

**Figure 8.** This figure demonstrates the data of the three regions that are used for the training, validating, and testing of the super-resolution algorithms. The visualization is based on EeteS simulated Sentinel-2 image. The background, testing area, validating area, and training area are visualized by R-G-B (663 nm-560 nm-492 nm), IR-G-B (842 nm-560 nm-492 nm)), IR (842 nm), and IR (842 nm), respectively. IR is short for infrared.

## 3 Experiments

### 3.1 Super-resolution

#### 3.1.1 Selected algorithms

In this paper, we select representative super-resolution algorithms among the literature and demonstrate their performance on our MDAS data set. The experiment results can be treated as the baseline for further studies. Our selection of algorithms are based on two papers, Yokoya et al. (2017) and Zhang et al. (2020c), as both studies carry out comprehensive comparative experiments and publish their source codes. Yokoya et al. (2017) implements 10 conventional algorithms, tests them on 8 data sets, and evaluates the performance with 4 metrics. Based on the 32 evaluations (4 metrics on 8 data sets) in Yokoya et al. (2017), we selected HySure (Simoes et al. 2014) and CNMF (Yokoya et al. 2011), since they most frequently rank the first place, respectively, 13 and 8 times. Zhang et al. (2020c) implements 6 deep learning models, tests on 6 data sets, and applies 4 evaluation metrics. We choose SSR-NET (Zhang et al. 2020c) and ResTFNet (Liu et al. 2020) as they rank the first place for 18 and 6 times. In the following, we briefly recap the selected algorithms. CNMF (Yokoya et al. 2011) applies non-negative matrix factorization on both hyperspectral and multispectral images so that hyperspectral endmembers and multispectral abundance map are extracted and further fused to achieve an image of high resolution in both spectral and spatial domain. HySure (Simoes et al. 2014) attempts to fuse hyperspectral and multispectral images by fitting both of them to a latent representation. To do so, they optimize two quadratic terms that fit the two images to the latent representation and a regularizer to preserve edges. Liu et al. (2020) proposes a two-stream CNN whose two branches extract representation from hyperspectral and multispectral images respectively, fuse high level representations at a late stage, and recover an image of high spectral and spatial resolution with a reconstruction loss. Zhang et al. (2020c) trains a CNN to accomplish super-resolution by a loss function of three terms.

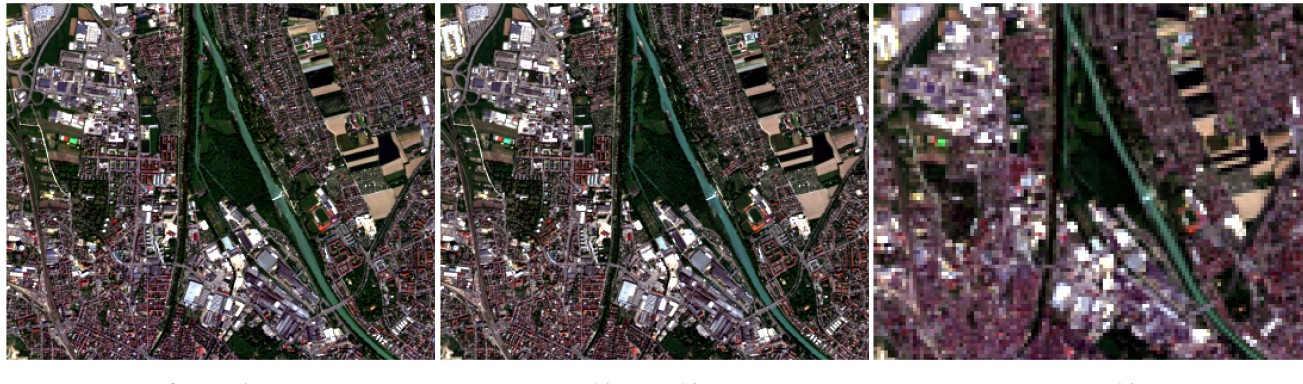

| Reference image | Multispectral image | Hyperspectral image |

**Figure 9.** Visualizations of reference image, multispectral image, and hyperspectral image for the testing area.

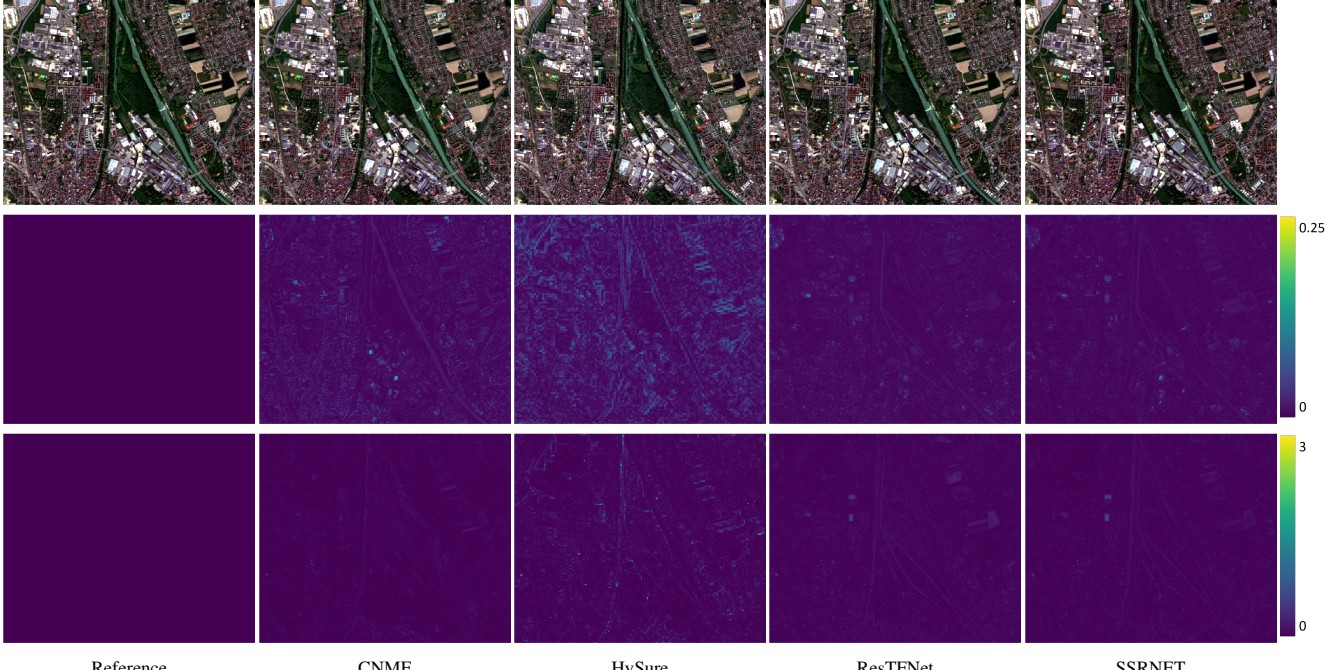

| Reference | CNMF | HySure | ResTFNet | SSRNET |

**Figure 10.** Visualizations of super-resolution results. The first row demonstrates RGB images composing of bands of 652 nm, 553 nm, and 450 nm. The second row demonstrates the rooted squared error between the reference and fusion results. The third row demonstrates the angle differences between the reference and fusion results.

These terms represent differences on spectral edges, spatial edges, and the data. Their training applies corresponding image patches of hyperspectral image, multispectral image, and the targeted image of high spectral and spatial resolution.





**Table 2.** Quantitative assessment of super-resolution algorithms

|  | PSNR | SAM | ERGAS | $Q2^n$ |
|---|---|---|---|---|
| CNMF (Yokoya et al. 2011) | 32.8202 | 4.3695 | 5.7793 | 0.5242 |
| HySure (Simoes et al. 2014) | 29.1428 | 7.4784 | 8.2372 | 0.4245 |
| SSR-NET (Zhang et al. 2020c) | **37.5386** | 2.9909 | 3.7842 | 0.6540 |
| ResTFNet (Liu et al. 2020) | 37.0841 | **2.9163** | **3.6529** | **0.6763** |

### 3.1.2 Evaluation metrics

For quantitative evaluations, we implement four metrics to assess the super-resolution results, namely, *PSNR*, *SAM*, *ERGAS*,
and *$Q2^n$*. *PSNR* evaluates the spatial reconstruction of each band. We assess our experiments with an averaged *PSNR* over all
bands. *SAM* is widely used to demonstrate the quality of spectral information via calculating the angle differences. *ERGAS* is
a modified version of mean squared error that globally assess the quality of data reconstruction. At last, *$Q2^n$* aims to qualify
spectral and spatial distortions. For more details about the four metrics, please refer to Yokoya et al. (2017).

### 3.1.3 Experiment setting

To train the algorithms, as shown in Fig. 8, we select three areas for training, validating, and testing the algorithms. This setting
is required by the two deep learning models, SSR-NET and ResTFNet. Since HySure and CNMF do not require training with
ground reference data, they work directly with data of the testing area.

On the aspect of data, we utilize the spatially simulated 10 m GSD hyperspectral image as the reference data, which have
high spatial and high spectral resolution. The low spatial and high spectral resolution image is the spatially simulated 30 m
GSD hyperspectral image. The high spatial and low spectral resolution image is the spectrally simulated 10 m multispectral
image. Fig. 9 visualizes these three images, and the details of simulations are introduced in Section 2.3.

The implementation of the deep learning models is based on the codes introduced in Zhang et al. (2020c), within PyTorch
framework. The number of epochs is 10,000. The optimizer is Adam with a learning rate of $10^{-4}$. The best model is saved
along the training procedure, based on the assessment on the validation data after each epoch. The codes of the shallow models
are accessible with Yokoya et al. (2017).

### 3.1.4 Experiment results and visualization

Table 2 shows the quantitative measures for the performance of the selected four super-resolution algorithms. ResTFNet has the
best performance among them by ranking the first for three indications. The two deep learning models significantly outperform
the other two conventional algorithms, at the cost of requiring dramatically more data for training models. Fig. 10 visualizes the
super-resolution results, images of root squared error, and images of angle differences. For both root squared error and angle
differences, HySure has the largest error. The two deep learning models have smaller errors compared to the two conventional
algorithms. It is interesting to point out that, in terms of angle differences, CNMF seems to outperform the two deep models.

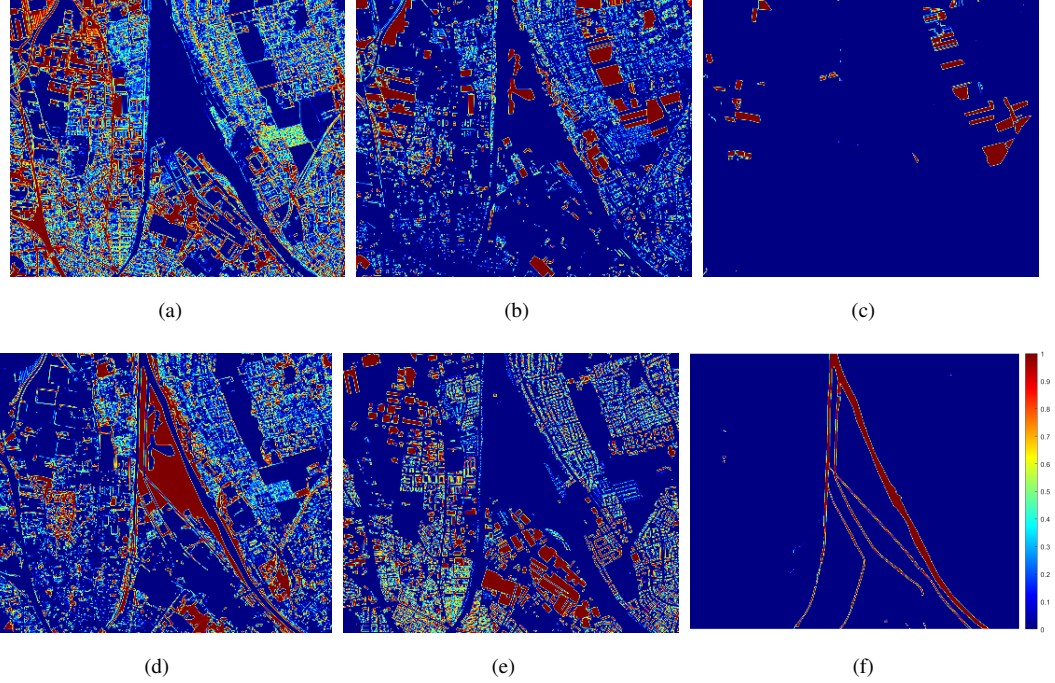

**Figure 11.** Ground truth of abundance maps of the sub1 region of the EeteS_EnMAP_10 data. (a) Pavement. (b) Low vegetation. (c) Soil. (d) Tree. (e) Roof. (f) Water.

## 3.2 Hyperspectral Unmixing

### 3.2.1 Ground truth

In the experiments, the endmember spectra shown in Fig. 4 were used by methods which need a spectral library as prior knowledge. The abundance ground truth for quantitative evaluation of unmixing performance was generated from the land cover map shown in Fig. 5-7. The generated abundance maps of the three subsets are displayed in Fig. 11-13.

### 3.2.2 Selected algorithms

There are several categories of unmixing algorithms (Hong et al. 2021a). In this paper, we tested the performance of algorithms
on the generated dataset that belong to these three categories: 1) Unmixing methods that don't need the spectral library as prior knowledge and don't consider the intra-class spectral variability. 2) Unmixing methods that don't need the spectral library as prior knowledge and consider the intra-class spectral variability. 3) Assume that there is a spectral library of pure spectra representing endmember variability for a set of endmember classes, and unmix the image with the library. The NMF-QMV algorithm (Zhuang et al. 2019), SeCoDe algorithm (Yao et al. 2021) and GMM algorithm (Zhou et al. 2020) are selected as
the representative algorithms that belong to the three categories, respectively. NMF-QMV conduct unmixing via introducing



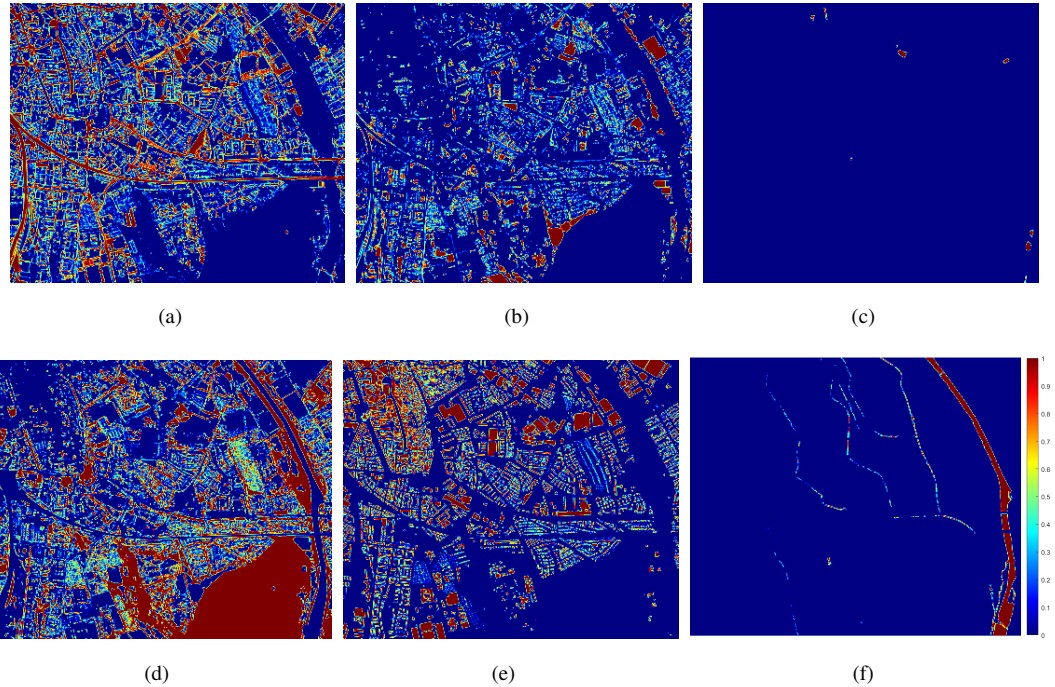

**Figure 12.** Ground truth of abundance maps of the sub2 region of the EeteS_EnMAP_10 data. (a) Pavement. (b) Low vegetation. (c) Soil. (d) Tree. (e) Roof. (f) Water.

three kind of well-known minimum volume regularization terms, referring to as *boundary*, *center* and *total variance* (TV) into the nonnegative matrix factorization framework, in which the regularization parameter between the data-fitting term and the regularization term is selected automatically. SeCoDe jointly capture the spatial-spectral information of hyperspectral image in a tensor-based fashion, in which the convolutional operation is employed to locally model the spatial relation between neigh-

310 boring pixels by spectral bundles that representing spectral variability (Hong et al. 2019a). Experiments on three hyperspectral datasets showed that SeCoDe outperformed other state-of-the-art unmixing algorithms. GMM assumes that the endmembers for each pixel are sampled from probability distributions, hence the pixels as linear combinations of these endmembers also follow the distribution. It works by using Gaussion mixture model for distributions and unmixing pixels based on the estimated distribution parameters. Experiments on both synthetic and real hyperspectral images showed that GMM performed best among

315 the distribution-based methods and achieved comparable unmixing accuracy to set-based methods without the need of library reduction, it may also be more stable across datasets.

### 3.2.3 Evaluation metric

The spectral angle distance (SAD) and root-mean-square-error (RMSE) are two most widely used evaluation metrics for unmixing (Liu and Zhu 2020). Since the tested methods contain the method that uses spectral library as prior information, only

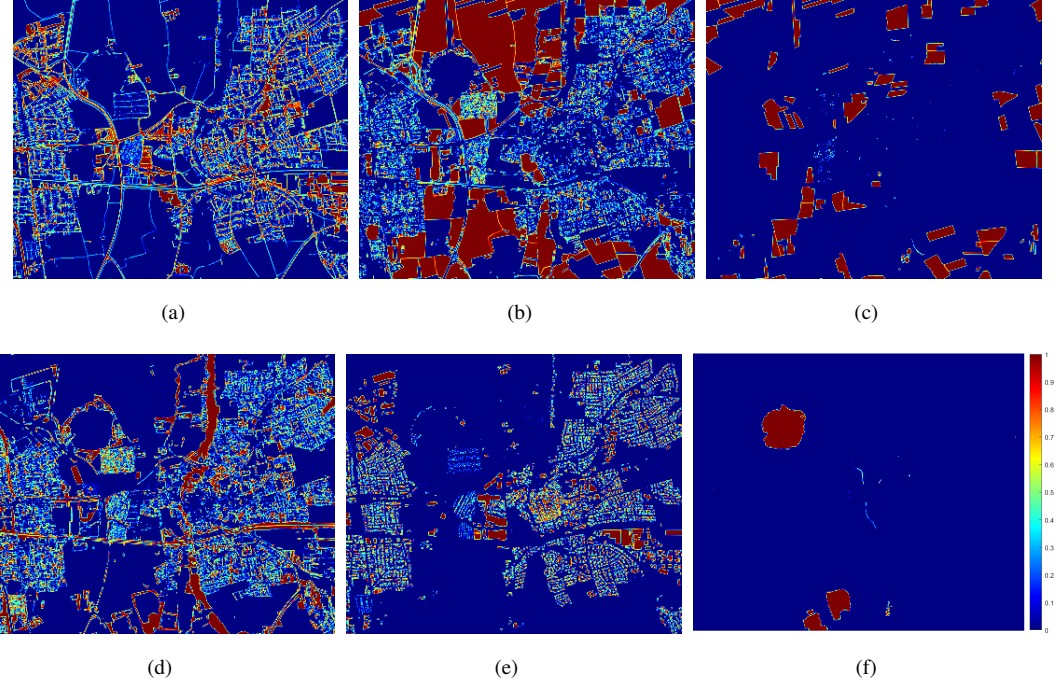

**Figure 13.** Ground truth of abundance maps of the sub3 region of the EeteS_EnMAP_10 data. (a) Pavement. (b) Low vegetation. (c) Soil. (d) Tree. (e) Roof. (f) Water.

RMSE between the estimated abundance and the ground truth abundance is employed for quantitative evaluation. A smaller RMSE value means better performance.

### 3.2.4 Experiment setting

We tested the selected algorithms on the three subsets of HySpex data and the EeteS_EnMAP_10 data. For NMF-QMV, the three regularization terms, *boundary*, *center* and TV are chosen, separately, and no other parameter needs to set. For SeCoDe,

there are three regularization parameters ($\alpha, \beta$ and $\gamma$) that need to be manually adjusted. We conducted parameter analysis on the sub1 region of EeteS_EnMAP_10 data and the parameter combination that produce the minimum mean RMSE value is $\alpha = 0.3$, $\beta = 0.01$, and $\gamma = 3$. This parameter combination was directly used for other images. For GMM, no parameter was manually set. All the algorithms were implemented in Matlab on a PC with Intel Core i7-6700 CPU and 64 GB memory.

### 3.2.5 Results

Table 3-5 show the quantitative assessments for the performance of the selected unmixing algorithms. For both of the HySpex and EnMAP dataset, SeCoDe has the best performance in terms of RMSE. Regarding the computation time, although SeCode is the most efficient, it needs extra time to select the best parameters while the NMF-QMV based methods don't need parameter



**Table 3.** Quantitative Assessment of Unmixing Algorithms for the Sub1 Region

|  |  | SeCoDe | NMF-QMV-Boundary | NMF-QMV-Center | NMF-QMV-TV | GMM |
|---|---|---|---|---|---|---|
| RMSE | HySpex | 0.3090 | 0.3291 | 0.3654 | 0.3211 | 0.3578 |
|  | EeteS_EnMAP_10 | 0.2560 | 0.2792 | 0.2777 | 0.2776 | 0.2873 |
| Time (min) | HySpex | 294.76 | 308.15 | 303.38 | 306.34 | 6442.33 |
|  | EeteS_EnMAP_10 | 8.76 | 14.23 | 14.65 | 14.94 | 194.3 |

**Table 4.** Quantitative Assessment of Unmixing Algorithms for the Sub2 Region

|  |  | SeCoDe | NMF-QMV-Boundary | NMF-QMV-Center | NMF-QMV-TV | GMM |
|---|---|---|---|---|---|---|
| RMSE | HySpex | 0.2786 | 0.2825 | 0.2899 | 0.2889 | 0.3249 |
|  | EeteS_EnMAP_10 | 0.2413 | 0.2557 | 0.2785 | 0.2772 | 0.2639 |
| Time (min) | HySpex | 287.06 | 274.52 | 279.53 | 277.50 | 5050.08 |
|  | EeteS_EnMAP_10 | 7.10 | 13.97 | 13.88 | 13.87 | 162.77 |

tuning. GMM is the most time-consuming method. The efficiency of all methods decreased largely when the size of the image increased.

## 3.3 Multimodal Land Cover Classification

### 3.3.1 Experimental setup

We selected pixel-wise classification as a potential application to evaluate the performance of different modality combinations for the land cover mapping task on the Augsburg dataset. The classification performance is quantitatively shown in terms of three common indices, i.e., *Overall Accuracy (OA)*, *Average Accuracy (AA)*, and *Kappa Coefficient ($\kappa$)* using canonical correlations forest (CCF) classifier (Rainforth and Wood 2015). We choose the sub area same as Fig. 5 to conduct our experiments. The sample statistics are summarized in Table. 6, in which 20% samples were randomly selected as training set.

**Table 5.** Quantitative Assessment of Unmixing Algorithms for the Sub3 Region

|  |  | SeCoDe | NMF-QMV-Boundary | NMF-QMV-Center | NMF-QMV-TV | GMM |
|---|---|---|---|---|---|---|
| RMSE | HySpex | 0.3185 | 0.3314 | 0.3484 | 0.3493 | 0.3400 |
|  | EeteS_EnMAP_10 | 0.2680 | 0.2997 | 0.2977 | 0.2973 | 0.3005 |
| Time (min) | HySpex | 240.40 | 294.70 | 303.64 | 272.32 | 6238.65 |
|  | EeteS_EnMAP_10 | 6.82 | 14.09 | 14.07 | 14.03 | 156.42 |





**Table 6.** This table lists category information of the investigated dataset.

| Class No. | Class Name | Training | Testing |
|:---:|:---:|:---:|:---:|
| 1 | Forest | 1235 | 4936 |
| 2 | Park | 55 | 220 |
| 3 | Residential | 8704 | 34812 |
| 4 | Industrial | 2487 | 9947 |
| 5 | Farm | 1456 | 5821 |
| 6 | Cemetery | 321 | 1284 |
| 7 | Allotments | 866 | 3460 |
| 8 | Meadow | 256 | 1023 |
| 9 | Commercial | 950 | 3797 |
| 10 | Recreation Ground | 598 | 2389 |
| 11 | Retail | 340 | 1358 |
| 12 | Scrub | 236 | 941 |
| 13 | Grass | 72 | 285 |
| 14 | Heath | 36 | 141 |
| - | Total | 17612 | 90388 |

In this task, we investigate the classification performance with the use of different combinations of RS modalities. They include

- single modality, e.g., HS, MS, SAR, DSM;

- two modalities, e.g., HS-MS, HS-SAR, HS-DSM, MS-SAR, MS-DSM, SAR-DSM;

- three modalities, e.g., HS-MS-SAR, HS-MS-DSM, HS-SAR-DSM, MS-SAR-DSM;

- four modalities, e.g., HS-MS-SAR-DSM.

### 3.3.2 Results and Discussion

Table 7 lists the classification results of all possible combinations of modalities on the to-be-released multimodal benchmark dataset, i.e., Augsburg. Moreover, only using HS or MS data is apt to obtain better classification results than SAR or DSM. The input with two modalities (even though SAR plus DSM) obviously yields a performance improvement compared to the single modality. Similarly, the use of three modalities achieves a better result on the basis of two modalities. As expected, the classification accuracies by using all modalities are superior to those by using single modality, two modalities, and three modalities, achieving the best classification performance. This can demonstrate the necessity and important values to investigate and build such a multimodal benchmark dataset in the RS community, which can help us break the performance bottleneck in many single-modality-oriented applications and meanwhile point out the future research direction.





**Table 7.** Classification results of different modality combinations using the CCF classifier on the investigated dataset.

| Class No. | 1 | 2 | 3 | 4 | 5 | 6 | 7 | 8 | 9 | 10 | 11 | 12 | 13 | 14 | OA (%) | AA (%) | Kappa |
|---|---|---|---|---|---|---|---|---|---|---|---|---|---|---|---|---|---|
| Single Modality | | | | | | | | | | | | | | | | | |
| HSI | 83.65 | 6.82 | 95.93 | 63.08 | 89.13 | 72.2 | 23.96 | 59.82 | 44.93 | 61.74 | 38.66 | 41.02 | 9.12 | 35.46 | 51.82 | 78.77 | 0.6817 |
| MSI | 84.14 | 9.55 | 94.85 | **73.97** | 83.51 | 38.86 | 48.99 | 61.00 | 41.59 | 62.45 | 29.31 | 22.21 | 11.58 | 15.60 | 48.40 | 79.48 | 0.6988 |
| SAR | 6.02 | 0.00 | 82.33 | 22.93 | 61.66 | 1.87 | 5.55 | 2.83 | 7.51 | 1.88 | 1.33 | 1.91 | 0.35 | 0.71 | 14.06 | 50.33 | 0.2055 |
| DSM | 17.54 | 1.82 | 51.67 | 19.34 | 26.39 | 3.50 | 9.74 | 3.23 | 7.93 | 3.77 | 3.31 | 2.66 | 0.35 | 0.71 | 10.85 | 32.94 | 0.0640 |
| Two Modalities | | | | | | | | | | | | | | | | | |
| HSI+MSI | 85.03 | 8.18 | 96.88 | 67.87 | 89.97 | 74.69 | 38.84 | 59.73 | 47.09 | 67.06 | 38.95 | 45.59 | 7.72 | 39.01 | 54.76 | 81.33 | 0.7216 |
| HSI+SAR | 84.26 | 7.27 | 95.94 | 63.23 | 88.71 | 72.04 | 26.04 | 58.55 | 44.98 | 62.12 | 38.22 | 42.30 | 9.47 | 37.59 | 52.19 | 79.02 | 0.6843 |
| HSI+DSM | 84.16 | 6.36 | 96.11 | 63.09 | 88.95 | 72.90 | 25.17 | 58.36 | 45.22 | 63.08 | 37.78 | 43.25 | 9.82 | 33.33 | 51.97 | 79.11 | 0.6853 |
| MSI+SAR | 85.09 | 5.00 | 95.15 | 75.12 | 85.86 | 38.47 | 50.84 | 57.77 | 43.64 | 64.46 | 32.11 | 20.40 | 9.47 | 12.06 | 48.25 | 80.26 | 0.7101 |
| MSI+DSM | 85.56 | 10.91 | 94.90 | 73.74 | 84.81 | 46.18 | **52.89** | **61.88** | **50.22** | 63.50 | 36.52 | 30.18 | **12.98** | 24.82 | 52.08 | 80.79 | 0.7191 |
| SAR+DSM | 29.96 | 8.18 | 85.87 | 25.84 | 68.65 | 5.14 | 10.03 | 12.12 | 12.88 | 3.47 | 2.21 | 0.64 | 1.75 | 1.42 | 19.15 | 55.54 | 0.2859 |
| Three Modalities | | | | | | | | | | | | | | | | | |
| HSI+MSI+SAR | 85.17 | 8.64 | **97.06** | 68.12 | 89.66 | 75.23 | 39.25 | 61.78 | 47.83 | 67.94 | 38.81 | 46.97 | 9.12 | 41.84 | 55.53 | 81.60 | 0.7255 |
| HSI+MSI+DSM | **85.64** | 8.18 | 96.99 | 68.43 | 89.83 | **76.09** | 38.64 | 59.14 | 48.56 | 67.43 | 39.79 | 45.59 | 9.12 | 43.26 | 55.48 | 81.62 | 0.7259 |
| HSI+SAR+DSM | 83.95 | 6.82 | 96.12 | 63.43 | 88.75 | 70.02 | 26.39 | 59.43 | 46.22 | 63.16 | 39.47 | 42.30 | 9.82 | 34.04 | 52.14 | 79.23 | 0.6874 |
| MSI+SAR+DSM | 85.74 | **12.73** | 95.40 | 73.56 | 87.01 | 45.48 | **52.89** | 59.53 | 49.26 | 63.71 | 36.67 | 28.16 | 12.28 | 24.11 | 51.89 | 81.09 | 0.7228 |
| Four Modalities | | | | | | | | | | | | | | | | | |
| HSI+MSI+SAR+DSM | 82.84 | 7.73 | 97.01 | 68.74 | **90.53** | 75.93 | 40.26 | 61.29 | 49.22 | **69.02** | **40.57** | **50.05** | 10.53 | **43.97** | **56.48** | **82.03** | **0.7324** |

**Figure 14.** Visualization of classification results obtained by the CCF classifier using different modalities.

Here, we only show the baseline results as a reference by using the CCF classifier directly conducted on the original RS data in order to provide a larger space to the subsequent potential researchers who are interested in the topics related to multimodal feature extraction and classification for RS data. In addition, we also visualize the corresponding classification

maps, as illustrated in Fig. 14, where there is a basically similar trend with quantitative results.



## 4 Conclusions

In this paper, we contribute to the community a new multimodal benchmark data set, MDAS, at the scene of Augsburg, Germany. It includes five different data modalities, which are SAR, multispectral image, hyperspectral image, DSM, and GIS data. The data sets are collected on the same day, $7^{th}$ May 2018. Experienced experts put a lot of efforts on the collections and preparations to ensure a high quality. MDAS not only provide a new benchmark data for current data fusion applications, but some of its components enrich data options for applications of single data source, such as the DSM and hyperspectral images. Additionally, the multiple modalities make it possible to explore different combinations. Besides the data itself, we contribute by demonstrating the performance of state-of-the-art algorithms on MDAS in terms of super-resolution, unmixing, and land cover classification. These experiment results provide baselines for further studies that uses MDAS.

## 5 Code and data availability

The data is accesible at https://mediatum.ub.tum.de/1657312 with a CC BY-SA 4.0 license ( Hu et al. (2022)), and the code (including the pre-trained models) at https://github.com/zhu-xlab/augsburg_Multimodal_Data_Set_MDaS

*Author contributions.* The dataset was conceptualized by XZ, curated by JH, RL, DH, technically supported by JY, MS, FK and KS. AC contributed to project administration. JH, RL and DH prepared the original draft, and all others revised it.

*Competing interests.* The authors declare that they have no conflict of interest.

*Acknowledgements.* It is impossible to accomplish this work without the help of a lot of colleagues. We would like to thank Ms. Chaonan Ji, Dr. Marianne Jilge, and Dr. Uta Heiden for their inspiring discussions, thank Dr. Martin Bachmann and Ms. Stefanie Holzwarth for sharing us their experience on preparing hyperspectral images, last but not the least, Dr. Rudolf Richter for supporting us on the software ATCOR. The work of X. Zhu is jointly supported by German Federal Ministry of Economics and Technology in the framework of the "national center of excellence ML4Earth" (grant number: 50EE2201C), the Helmholtz Association through the framework of Helmholtz AI (grant number: ZT-I-PF-5-01) - Local Unit "Munich Unit @Aeronautics, Space and Transport (MASTr)" and Helmholtz Excellent Professorship "Data Science in Earth Observation - Big Data Fusion for Urban Research"(grant number: W2-W3-100), by the German Federal Ministry of Education and Research (BMBF) in the framework of the international future AI lab "AI4EO – Artificial Intelligence for Earth Observation: Reasoning, Uncertainties, Ethics and Beyond" (grant number: 01DD20001).



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
