# Peer review of "MDAS: A New Multimodal Benchmark Dataset for Remote Sensing"

_Earth System Science Data, 2022_

## Author Response (AR1)

**Referee #1**

In the manuscript a multimodal dataset (MDAS) is presented, and experiments were carried out with state of the art methods as a benchmark. The contributions are stated clearly, and the methodologies that were employed explained in detail and correctly referenced.

Despite focusing only on a restricted area on a single date, MDAS can provide the community with a dataset  useful for the development and testing of data fusion approaches, with a solid benchmark already in place.

Dear Reviewer, thank you very much for your positive comments and review. Please, find the response to your comments below:

I have some minor comments:

* Is an extension to the dataset in terms of area and dates possible or even already foreseen?

An extension is possible. However, currently, we have no concrete plans to extend the data set (due to the high cost of data collection and preparation).

* Are you going to provide the code for all the benchmarking experiments in your repository? At the current state, I can only see those for super-resolution, and not for spectral unmixing and multimodal land cover classification

We have updated the repository (https://github.com/zhu-xlab/augsburg_Multimodal_Data_Set_MDaS) with links to the original implementations of the missing benchmarking algorithms.

A technical question:

* on pag. 15, line 283, is 10.000 the number of epochs or the number of iterations?

We have cross-checked the manuscript and the git repository, and 10,000 is the number of epochs (as it is stated on page 15, line 283).

**Referee #2**

In this work, the authors propose a new multimodal benchmark dataset for remote sensing. There are two main contributions: 1) a new multimodal benchmark data, named MDAS, consists of five modalities: SAR data, multispectral image, hyperspectral image, DSM, and GIS data; 2) three typical remote sensing applications in the MDAS dataset are conducted with the state-of-the-art (SOTA) algorithms. Apparently, the MDAS dataset is well-prepared by experienced experts with high quality, being useful for various applications. This paper is well written in general with clear motivations and nice illustrations.

Thank you very much for the provided feedback. We have updated the manuscript accordingly, and below you will find a detail answer to the issues.

Here are some suggestions for major modifications.

1. In the section 1-introduction, this paper has listed different datasets in the three typical fields to reflect the advantages of the MDAS dataset, but the MDAS is not limited by three applications. If a summary table of different datasets is attached in the section 1, it would be more intuitive to reflect the differences between the MDAS and other datasets. It is recommended to analyze from the perspective of data type, covering area, acquisition difficulty, etc.

We have included a new table (Table 1. Remote sensing data sets comparison…) to highlight the differences (and adavantages) between MDAS an the most popular data sets in the field. Namely, MDAS offers five data modalities, covering a whole city, with data collected on the same day, and professional preprocessing and simulations.

| Data | Modality | | | | | Resolution (m) | Task | No. of Images | Image size |
|---|---|---|---|---|---|---|---|---|---|
| | SAR | Multispectral | Hyperspectral | DSM | GIS | | | | |
| UC-Merced | | ✓ | | | | 0.3 | Image classification | 2100 | 256x256 |
| WHU-RS19 | | ✓ | | | | up to 0.5 | Image classification | 950 | 600x600 |
| AID | | ✓ | | | | within 10 | Image classification | 10000 | 600x600 |
| BigEarthNet | ✓ | ✓ | | | | 10, 10 | Multi-label classification | 590326 | 120x120 |
| Vaihingen | | ✓ | | ✓ | | 0.09x0.09 | Semantic segmentation | 38 | up to 2000x3000 |
| Potsdam | | ✓ | | ✓ | | 0.05x0.05 | Semantic segmentation | 38 | 6000x6000 |
| Houston | | ✓ | | ✓ | | 2.5x2.5 | Semantic segmentation | 1 | 349x1905 |
| So2Sat LCZ42 | ✓ | ✓ | | | | 10x10 | Image classification | 400673 | 32x32 |
| DOTA | | ✓ | | | | - | Object detection | 2806 | 4000x4000 |
| GID | | ✓ | | | | up to 0.5 | Semantic segmentation | 150 | 6800x7200 |
| SAR-Ship (SSDD) | ✓ | | | | | | Object detection | 39729 | 256x256 |
| DFC 2018 | | ✓ | ✓ | ✓ | | 0.05, 1, 0.5 | Image classification | 3 | 11920x12020, 4172x1202, 8344x2404 |
| Berlin-Urban-Gradient dataset 2009 | | | ✓ | | | 3.6 x 9 | Unmixing and classification | 2 | 6895x1830, 2722x732 |
| MDAS | ✓ | ✓ | ✓ | ✓ | ✓ | Check Table 2 for the details | | | |

2. In the section 2.1-synthetic aperture radar data, the information of SAR data needs to be introduced more detailly. The SAR data after processing is the backscattering coefficient or some other format? Is the SAR data range between 0 and 1 or is it converted to dB? Whether is the SAR data processed after speckle denoising? How to the SAR data in the experiments? Particularly, speckle noises have a great impact on remote sensing applications. These details need to be further introduced.

Thanks for highlighting this missing information. We have updated Section 2.1 accordingly. Particularly, the SAR data after processing is the backscattering coefficient. We didn't apply speckle denoising, and the data was not converted to dB. The preprocessing of the SAR data was conducted by using SNAP toolbox. We fetched a level-1 Ground Range Detected (GRD) product and applied precise orbit profile, conducting radiometric calibration, and terrain correction.

3. In the table 1, it is necessary to list the resolution and band number of all mentioned data and labels, because the MDAS involves a lot of data processed by software.

We have updated the referred table (now, Table 2).

| Data | Modality | Sensor | GSD (m) | No. Bands |
|---|---|---|---|---|
| Sentinel-1 | SAR | Sentinel-1 payload | 10 | 2 |
| Sentinel-2 | Multispectral image | Sentinel-2 payload | 10 | 12 |
| DSM | DSM | DLR 3K | 0.25 | 1 |
| HySpex | Hyperspectral image | HySpex | 2.2 | 416 |
| S2eteS_S2 | Multispectral image | S2eteS Spatial and spectral simulation | 10 | 4 |
| EeteS_EnMAP_10 | Hyperspectral image | EeteS Spatial and spectral simulation | 10 | 242 |
| EeteS_EnMAP_30 | Hyperspectral image | EeteS Spatial and spectral simulation | 30 | 242 |
| EnMAP | Hyperspectral image | EeteS | 30 | 242 |
| GIS | GIS | Open street map (OSM) | | |
| Endmember | | Manual labeling | | |
| Land cover maps | | Manual labeling | | |

4. In the section 3.1-super-resolution, the evaluation metrics have PSNR, and the optical data are all the BOA data, which range from 0 to 1. For the convenience of storage, the BOA data is always uint16 format from 0 to 10000. In the experiments of super-resolution, this paper used the original data from 0 to 1, or max-min normalize the 0-10000 data to the uint8 format from 0 to 255. If the original data of 0-1 is adopted, the PSNR will be large, and it is better to use RMSE.

Thanks for raising this issue. We have double checked the code (and the results). Particularly, when calculating PSNR, the original data and the superresolution results are all in the range of 0 to 10000. Therefore, the PSNR value informed should be consistent. On the other hand, the super resolution experiment results includes three aditional metrics. Thus, the readers should have additional information to judge the results.

5. In the section 3.3- multimodal land cover classification, this paper only compares the results of the same algorithm (2015) under different data input. This comparison is convincing. But the previous two fields are the results of some SOTA algorithms in recent years. This part needs to increase the comparison experiments of different SOTA algorithms in the multimodal land cover classification.

Our paper is related to opening the multimodal remote sensing data rather than the peformance comaprison of the SOTA methods, particularly for the classification problem. Moreover, around the 'multimodal' data, it would be better to see the performance gain with the use of different data combination. For a fair comparison, we use the same classifier. i.e., CCF.  For the tasks 1 and 2, it is only related to data fusion with hyperspectral and multispectral, spectral unmixing with only hyperspectral. It is natural to see the performance difference with different methods.

6. Readers may be interested in whether the MDAS dataset can be applied in some other directions in addition to the above-mentioned three applications. And the mentioned SOTA algorithms, especially the deep learning model trained on this dataset can be applied in the any other area. These issues should be discussed.

The reviewer is right, it is important to highlight some possible uses of our data sets beyond the three applications presented in the manuscript. Therefore, we have included a list of recommended directions in section *4. Conclusions*.

---

## Author Response (AR2)

**Topical Editor decision: Publish subject to technical corrections**

**Comments to the author:**
Thanks for the detailed response to the reviewers comments.

Dear Prof. Schultz, thanks for taking care of the review process.

I have 3 small things to ask you before the paper can be published:
1) please improve Figure 10 - the lower two rows appear as almost monochrome blue panels

Taking into account your advice, we have produced a new version of Figure 10. Hopefully, the new version will be easier to interpret.

2) in the code and data availability section please include the link to a permanent copy of the git repo (for example on zenodo). This snapshot copy may well include a link back to the live repo, but for reproducibility reasons it is important that the code is available at a location that is outside your control (once you published it there). You can include both links in the code and data availability section then.

We published the code on zenodo (https://doi.org/10.5281/zenodo.7428215). In the code and data availability section we have included the permanent DOI, and a link to the live repository, as suggested.

3) please run a spell checker one more time - I discovered a typo at the beginning of the Conclusions. Perhaps there are more?

Thanks for noticing this. We checked the entire manuscript, thus the new version should be OK (or at least have few typos).

Once again, thanks for managing the revision process.